# Video Creation by Demonstration

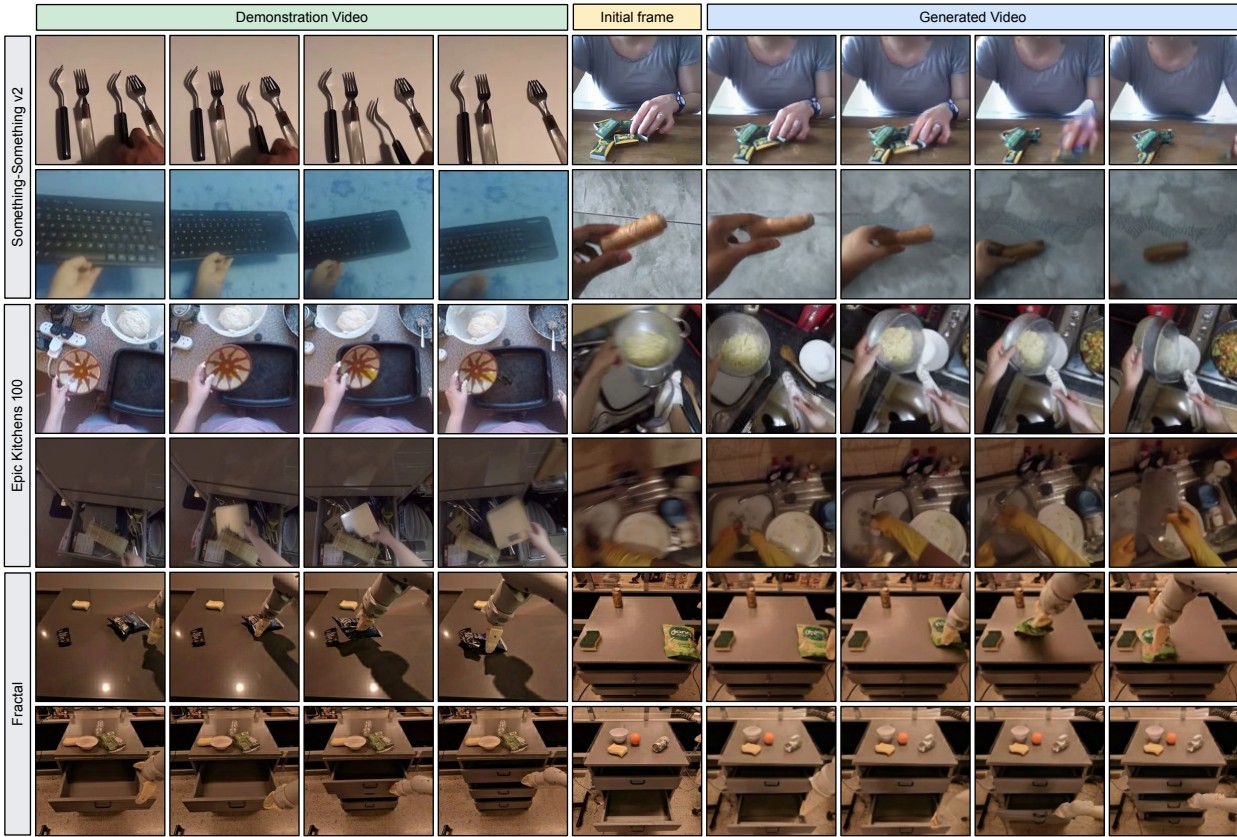

Figure 1: **Video Creation by Demonstration.** Given a demonstration video and an initial frame, our proposed $\delta$-Diffusion generates a video that naturally continues from the initial frame and carries out the same action as shown in demonstration.

## Abstract

We present Video Creation by Demonstration: given a demonstration video and an initial frame from any scene, we generate a realistic video that continues naturally from the initial frame and carries out the action concepts from the demonstration. This is important because unlike captions, camera poses, or point tracks, a demonstration video can provide detailed description of the target action without needing extensive manual annotations. The main challenge for training these models is the difficulty in curating supervised training data based on paired actions across different contexts. To mitigate this, we propose $\delta$-Diffusion, a self-supervised method that learns from unlabeled videos. Our key insight is that by placing a separately learned bottleneck on the features of a video foundation model, we can extract demonstration actions through these features and minimize degenerate solutions. We found $\delta$-Diffusion to outperform baselines in both human preference and large-scale machine evaluations.

# 1 Introduction

When given a visual demonstration, humans can naturally imagine what it would look like if these actions were to take place in a different environment. This leads to a natural question, can we teach machines to do the same? As an initial step towards answering this question, we propose *Video Creation by Demonstration*, a video creation experience that empowers users to generate videos by providing a demonstration video that showcases desired actions and an initial frame of the scene to carry out the actions from. As shown in Figure 1, our system can generate a new video that integrates the demonstrated action into the provided context, ensuring both temporal continuity and physical plausibility.

With the recent advances in diffusion models (Ho et al., 2020; Song et al., 2021), video generation (Brooks et al., 2024; Polyak et al., 2024; Gupta et al., 2024) emerges as a frontier in the goal of building interactive world simulators (Brooks et al., 2024; Bruce et al., 2024; Valevski et al., 2024; Alonso et al., 2024; Meng et al., 2024; Yang et al., 2024b). Compared to Video Creation by Demonstration, most existing video generation approaches typically rely on control signals that are either abstract by nature (*e.g.*, text prompts (Gupta et al., 2024), moving keypoints (Wu et al., 2024)) or difficult to acquire (*e.g.*, dense depth (Chen et al., 2023c) or segmentation maps (Han et al., 2022)). In contrast, a demonstration video can describe the target action in details without needing extensive manual annotations.

Unlike domain-specific approaches (Bruce et al., 2024; Valevski et al., 2024; Song et al., 2019; Siarohin et al., 2021), our work targets general videos, which present significant challenges as actions are naturally contextualized and highly complex. Consider Figure 1 row 5, the same action concept of "moving object left" can appear drastically different depending on the subject performing the action (*e.g.*, different trajectories of the robot arm), the object being acted upon (*e.g.*, different identities and poses), and the surrounding environment (*e.g.*, possible collisions with other objects). This misalignment, along with inherent complexities in videos such as camera view changes and motion blurs, poses significant challenges for transferring action concepts between different contexts. On one hand, this fundamentally differentiates our work from conventional action transfer and retargeting (Siarohin et al., 2019; 2021; Ren et al., 2021), which typically assume strict alignment between both actions and contexts of reference and target scenes. On the other hand, this misalignment prevents us from mining paired training data as in previous methods (Siarohin et al., 2021; Ren et al., 2021), making it challenging for model training.

To address these challenges, we propose $\delta$-Diffusion. First, we leverage video foundation models (Wang et al., 2022; Zhao et al., 2024a; Wang et al., 2024c; Bardes et al., 2024) to effectively encode complex actions in the demonstration videos. Second, we adopt self-supervised training to alleviate the need for supervised paired data. From a single video, we sample an initial frame and its following frames as demonstration to guarantee context-action alignment, and task the model to generate the same demonstration video. However, this would naturally lead to degenerate solutions where the initial frame is ignored during inference.

To mitigate this issue, we adopt a two-stage training paradigm. First, by learning to predict the spatiotemporal features from individual frames independently, we formulate an appearance bottleneck that preserves action signals and eliminates appearance information in the demonstration video features. Second, we train a diffusion model to predict the demonstration video by conditioning on its bottlenecked action representation and the initial frame. With minimal appearance information in the action representation, the model can only rely on the initial frame for the appearance of the generation target. With these key designs, $\delta$-Diffusion can generate videos with realistic motion that seamlessly integrates with the specified action and context.

We demonstrate the effectiveness of our method in terms of visual quality, action transferability, and initial frame consistency through both machine and human evaluations on diverse video datasets. Notably, $\delta$-Diffusion is capable of generating high-fidelity videos spanning a wide range of action concepts, from everyday activities and ego-centric perspectives to complex robotic actions. We also show that creating videos by visual demonstration yields better controllability and concept transferability compared to text control. Furthermore, we are able to use different demonstration videos simply concatenated together to drive the generation of a coherent sequence, indicating the potential of leveraging $\delta$-Diffusion as an alternative to generative interactive environment (Bruce et al., 2024).

In summary, we make the following contributions. (i) We introduce Video Creation by Demonstration, a new creation experience for controllable video generation, which enables users to directly use videos as driving control signals for transferring action concepts. (ii) To the best of our knowledge, we are the first to leverage out-of-the-box video foundation models for latent control of video generation. (iii) We propose a novel self-supervised approach for model training, which achieves compelling controllable video generation results. We hope the proposed paradigm for controllable video generation will open new doors to interactive world simulation.

## 2 Related Works

### 2.1 Video Generation

Recent years have witnessed significant progress in video generation (Blattmann et al., 2023; Harvey et al., 2022; Ho et al., 2022a;b; Singer et al., 2023; Brooks et al., 2024; Polyak et al., 2024; Gupta et al., 2024), with a range of methods for controlling the generation process, including text-to-video (Ramesh et al., 2021; Ho et al., 2022a; Singer et al., 2023), image-to-video (Zhao et al., 2018; Singer et al., 2023), image+text-to-video (Gupta et al., 2024; Xiang et al., 2024; Wang et al., 2024d), and image+video-to-video (I+V2V) (Zhao et al., 2024b). Our work, Video Creation by Demonstration, falls into the I+V2V category, where we aim to generate a video that continues from the initial frame while integrating the action concepts from the demonstration video.

Existing I+V2V techniques typically extract either explicit or implicit control signals from the condition video, and use those to animate the input initial frame. The explicit signals include text prompts (Wang et al., 2024b; Yang et al., 2024a; Xiang et al., 2024; Hu et al., 2023; Yang et al., 2024c; Xing et al., 2024; Kong et al., 2024; Wan et al., 2025), depth maps and edge maps (Wang et al., 2024b; Chen et al., 2023c), box or point tracks (Li et al., 2024b; Chen et al., 2023b; Wu et al., 2024; Wang et al., 2024d;a; Gu et al., 2025; Geng et al., 2025; Namekata et al., 2024), human keypoints (Hu, 2024; Park et al., 2024; Li et al., 2024a), camera trajectories (Xu et al., 2024; Xing et al., 2025), segmentation masks (Xiao et al., 2024; Huang et al., 2022; Davtyan & Favaro, 2022), or a combination of multiple signals (*e.g.*, sketch, depth, style) (Wang et al., 2024b; Chen et al., 2023c). However, when there is no strict alignment between the initial frame and the condition video, the explicit signals extracted from condition video would be misaligned as well.

In comparison, implicit signal (*e.g.*, learned embeddings) would serve as a better control signal across the misalignment. Genie (Bruce et al., 2024) learns re-usable latent action codebook from video game videos, which can be extracted from demonstration video to control generation. In comparison, $\delta$-Diffusion extracts and utilize action concepts that are more abstract than consecutive frame changes, while applying to the real-world visual domain. In addition, MotionDirector (Zhao et al., 2024b) fine-tunes a video diffusion model for each demonstration video at test-time to generate the reference motion pattern that contains textures consistent with a given image. $\delta$-Diffusion, in contrast, aims to generate videos that naturally continue from the given initial frame, with no optimization during inference. Similar to $\delta$-Diffusion, Vid-ICL (Zhang et al., 2024) also predicts the future frames guided from videos. However, Vid-ICL requires a separate video query as input whereas we only assume a single initial frame, which offers less contextual information and greater range of feasible predictions.

### 2.2 Modeling Motion and Action

A key challenge in our framework is to effectively capture the action concept from the demonstration video, and transfer it to a different context given by the initial frame.

One line of work studies a related task of action transfer or action re-targeting (Song et al., 2019; Siarohin et al., 2019; 2021; Ren et al., 2021), which involves decomposing videos into motion and content representation, then transferring learned motion from one video into the content of another. While showing promising results, they often assume a high degree of alignment between source and target videos. This alignment (*e.g.*, humans facing camera at the same distance) does not hold for Video Creation by Demonstration, where the demonstration video and the initial frame can be misaligned. For instance, a video of a robot arm closing

the top open drawer can be paired with an initial frame where only a further away bottom drawer is open. In such cases, directly transferring low-level motion patterns may not accurately convey the intended action concept.

In addition, other works have investigated extracting action information for robotics applications. To model action between consecutive frames, DynaMo (Cui et al., 2025) uses latent inverse/forward dynamics models while LAPA (Ye et al., 2024) computes discrete action tokens. In contrast, we target general videos and model actions defined in much longer temporal ranges.

### 2.3 Video Foundation Models

The emergence of powerful video foundation models (Zhao et al., 2024a; Wang et al., 2022; 2024c; Bardes et al., 2024) has led to new advancements in video understanding. These models can extract rich representations that capture both action and context information, while maintain generalizability in a wide range of downstream tasks (Yuan et al., 2024). To the best of our knowledge, we are the first to leverage video foundation models to extract implicit action representation for controlled video generation.

## 3 Methodology

### 3.1 Task Formulation

For Video Creation by Demonstration, the input is an initial frame $I$ providing contextual information and a demonstration video $V$ providing the control signal for generation. The goal is to generate a video $\hat{V}$ that naturally continues from the initial frame $I$ and carries out the action concepts shown in the demonstration video $V$ (Figure 1). Furthermore, we only consider this task valid when the initial frame contains the objects necessary for supporting the demonstrated action. For instance, $I$ must contain a throwable object for a demonstration of "throwing a ball".

### 3.2 Method Overview

Figure 2(a) shows an overview of $\delta$-Diffusion. The generation model $\mathcal{G}$ takes an initial frame $I$ and control latents $\delta_V$ extracted from a demonstration $V$ as inputs, and outputs a desired video $\hat{V}$. To avoid costly curation of paired video data and improve scalability, we train $\delta$-Diffusion in a self-supervised manner on unlabeled videos only.

During training, we sample the input initial frame $I$ and demonstration $V$ from a single video, with $I$ being the starting frame followed by $V$. The model $\mathcal{G}$ is then tasked to reconstruct the same video $V$ as the target $\hat{V}$. Without a careful design, the model $\mathcal{G}$ may learn an identity mapping from demonstration video $V$ to target $\hat{V}$, ignoring initial frame $I$. To fix this, we propose a separately learned bottleneck that can extract action signals $\delta_V$ from $V$ with minimal appearance information. As $\delta_V$ limits the appearance leakage from demonstration $V$, the model $\mathcal{G}$ must rely on the initial frame $I$ to provide the appearance information.

### 3.3 Extracting Action Latents

To compute action latent $\delta_V$ from demonstration $V$, we first apply a pre-trained video encoder to extract $V$'s spatiotemporal representations $\mathbf{z}^{\mathrm{ST}} \in \mathbb{R}^{T \times N \times D}$, where $T$, $N$, and $D$ represent the temporal, spatial, and feature dimension, respectively. Naturally, $\mathbf{z}^{\mathrm{ST}}$ is entangled with both appearance and action information. To extract its action information, we build an appearance bottleneck from one *key insight*: unlike action information, which is defined by multiple consecutive frames, appearance information in $\mathbf{z}^{\mathrm{ST}}$ can be well-modeled by individual frames independently.

Figure 2(b) depicts the appearance bottleneck, which consists of a feature predictor $\mathcal{P}$ and a subtraction operation. Concretely, $\mathcal{P}$ takes the spatial features $\mathbf{z}_t^{\mathrm{S}} \in \mathbb{R}^{N \times D}$ from frame $V_t$ as input and outputs the best-effort approximation for the corresponding $t$-th frame slice of the spatiotemporal feature $\mathbf{z}_t^{\mathrm{ST}} \in \mathbb{R}^{N \times D}$. To learn $\mathcal{P}$, we directly minimize the difference between the spatiotemporal feature $\mathbf{z}_t^{\mathrm{ST}}$ and approximation

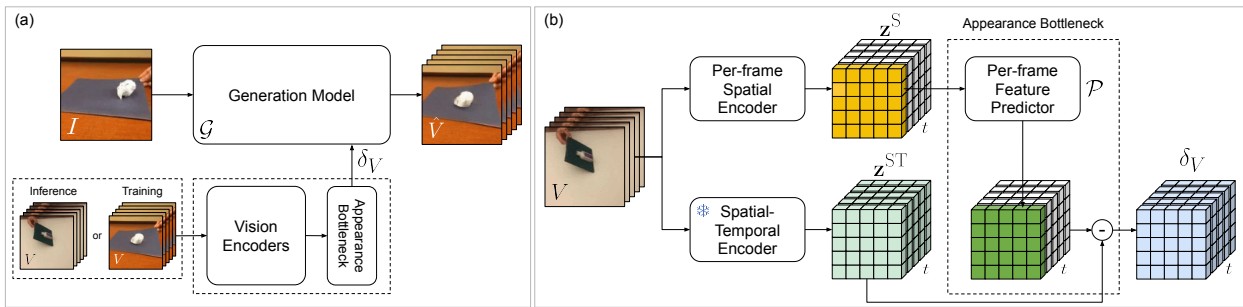

Figure 2: (a) **Overview of $\delta$-Diffusion.** The initial frame $I$ is provided to the generation model $\mathcal{G}$ along with the action latents $\delta_V$ extracted from the demonstration video $V$. The output $\hat{V}$ continues naturally from $I$ and carries out the actions in $V$. During training, the target $\hat{V}$ is the same as the demonstration $V$. (b) **Extracting action latents.** First, a video encoder extracts the spatiotemporal representations $\mathbf{z}^{\mathrm{ST}}$ from demonstration $V$, with $t$ denoting the temporal dimension. In parallel, an image encoder extracts per-frame spatial representations $\mathbf{z}^{\mathrm{S}}$ from $V$, which are aligned to $\mathbf{z}^{\mathrm{ST}}$ by feature predictor $\mathcal{P}$. The appearance bottleneck then computes the action latents $\delta_V$ by subtracting the aligned spatial representations $\mathcal{P}(\mathbf{z}_t^{\mathrm{S}})$ from the spatiotemporal representations $\mathbf{z}_t^{\mathrm{ST}}$ for each frame $V_t$.

$\mathcal{P}(\mathbf{z}_t^{\mathrm{S}})$ over the dataset $\mathcal{D}$:

$$\mathcal{P} = \arg\min_{\mathcal{P}} \sum_{V \in \mathcal{D}} \sum_{t=0}^{T-1} ||\mathbf{z}_t^{\mathrm{ST}} - \mathcal{P}(\mathbf{z}_t^{\mathrm{S}})||_2^2 \tag{1}$$

Intuitively, $\mathcal{P}$ is encouraged to estimate $\mathbf{z}_t^{\mathrm{ST}}$ as well as possible from the spatial feature $\mathbf{z}_t^{\mathrm{S}}$. However, without having access to the inter-frame "action" information, $\mathcal{P}$ can only capture the contextual appearance information in the demonstration $V$. Occasionally, $\mathcal{P}$ can also encode certain action information that is deducible from appearance alone. For instance, the concept of "cutting" can be inferred from an image of holding a knife over a cutting board. However, without adjacent frames, $\mathcal{P}$ cannot capture detailed action information such as speed/smoothness/direction of the cutting strokes.

Naturally, the difference between $\mathbf{z}_t^{\mathrm{ST}}$ and $\mathcal{P}(\mathbf{z}_t^{\mathrm{S}})$ captures a "temporal surprisal" and models the action information in $V$ retrievable only from multiple frames. We thus formulate the action control latents $\delta_V$ as follows where $[\cdot]$ is a concatenation operation.

$$\delta_V = \mathbf{z}^{\mathrm{ST}} - [\mathcal{P}(\mathbf{z}_0^{\mathrm{S}}), \mathcal{P}(\mathbf{z}_1^{\mathrm{S}}), \ldots, \mathcal{P}(\mathbf{z}_{T-1}^{\mathrm{S}})], \tag{2}$$

### 3.4 Training $\delta$-Diffusion

We then train the generation model $\mathcal{G}$ in a self-supervised manner. During training, we sample $T+1$ frames $v_{0:T}$ from a video in the training set, where $v_t$ denotes the $t$-th frame. The first frame $v_0$ is used as the initial frame $I$, and the subsequent frames $v_{1:T}$ are used both as the demonstration $V$ and the reconstruction target $\hat{V}$. In practice, we apply additional action-preserving augmentations (*e.g.*, random spatial cropping) to convert $V$ into $V'$ and compute $\delta_{V'}$. These augmentations further promote robustness to misalignment between inputs $I$ and $V$ and discourage $\mathcal{G}$ from directly copying $V$ during training.

Motivated by recent success of diffusion models (Brooks et al., 2024; Polyak et al., 2024; Gupta et al., 2024), we adopt latent diffusion models (LDMs) as $\mathcal{G}$. We first employ a tokenizer to compress both $I$ and $\hat{V}$ into a low dimensional latent space. The LDM then takes the corrupted latents of $\hat{V}$ together with other conditions as inputs and learns with a denoising loss function $\mathcal{L}$ in (Gupta et al., 2024).

$$\mathcal{G} = \arg\min_{\mathcal{G}} \sum_{v_{0:T} \in \mathcal{D}} \mathcal{L}\left(\mathcal{G}(I, \delta_{V'}), \hat{V}\right) \tag{3}$$

Following Gupta et al. (2024); Salimans & Ho (2022), we use velocity as the target for the denoising loss. During inference, $\mathcal{G}$ generates videos by iteratively denoising samples drawn from a noise distribution.

### 3.5 Implementation Details

We adopt VideoPrism (Zhao et al., 2024a) (Base, 0.1B parameters) as the vision encoders, which first computes the spatial features $\mathbf{z}^{S}$ and then temporally aggregates each spatial location to output the spatiotemporal feature $\mathbf{z}^{ST}$. For the appearance bottleneck feature predictor $\mathcal{P}$, we construct it as a stack of 4 Transformer encoder blocks from ViViT (Arnab et al., 2021). During training, $\mathcal{P}$ takes the spatial feature $\mathbf{z}_t^{S}$ for frame $V_t$ as input, and learns to predict the corresponding frame representations $\mathbf{z}_t^{ST}$ via L2 loss.

For the video generation model $\mathcal{G}$, we adopt WALT (Gupta et al., 2024) (Large, 0.3B parameters). $\mathcal{G}$ is trained to generate videos of $T = 16$ frames with $128 \times 128$ spatial resolution. The control latents $\delta_V$ from demonstration video $V$ is projected into a sequence of 2048-dimensional latent vectors in place of the original text embeddings for conditioning. Additional details are provided in the supplementary materials.

## 4 Experiments

### 4.1 Datasets

We conduct our experiments on three datasets, namely Epic Kitchens 100 (Damen et al., 2018), Something-Something v2 (SSv2) (Goyal et al., 2017), and Fractal (Brohan et al., 2022). We choose them for their rich human-object interactions and state changes in the video clips, making them a good testbed to demonstrate the proposed Video Creation by Demonstration. Something-Something v2 (SSv2) (Goyal et al., 2017) features a collection of videos showing 174 pre-defined actions with everyday objects. It contains both first-person-view and third-person-view videos with rich human-object interactions. Epic Kitchens 100 (Damen et al., 2018) is an egocentric kitchen video dataset, consists of 55 hours of videos with fine-grained action annotations. Lastly, Fractal (Brohan et al., 2022) is a real-world robotics manipulation dataset containing around 130k episodes over 700 tasks.

As Video Creation by Demonstration requires a pair of ⟨*initial frame, demonstration video*⟩ as input, trivially picking one image paired with a random video during inference may lead to incompatible action concepts. This is because the action in the demonstration video could be infeasible to execute in the initial frame. To mitigate this issue, we reorganize the existing datasets and curate meaningful pairs for validation. For each dataset, we randomly sample 1k video pairs within each of the top-10 most frequent verb categories. In each video pair $(A, B)$, we take video A as the demonstration video and the first frame from video B as the initial frame. The generated video is evaluated against video B as we assume the videos with the same labeled description contains transferable action concept. We use this curated large scale evaluation dataset for machine evaluation. Additionally, we select a set of video pairs by manually verifying the transferability of their action concepts. These selected video pairs are used for human evaluation and will be made available. Details of the selection process are found in supplementary.

### 4.2 Evaluation Setup

**Training Setup.** To train $\delta$-Diffusion, we first train a shared per-frame feature predictor $\mathcal{P}$ on a mixture of the training split of all the datasets mentioned above. We additionally include Ego4D (Grauman et al., 2022) during this training stage due to its rich and diversified content. Then, we freeze $\mathcal{P}$ and train the generation model $\mathcal{G}$ on each of the three datasets individually, yielding one model for each dataset.

**Baselines.** Other than video, text is also a natural format to describe actions. In our study, we take WALT (Gupta et al., 2024) (Large, 0.3B) as the representative of video generation conditioned on image and text. For a fair comparison, we fully fine-tune WALT on each dataset individually with ground truth captions. In addition, we compare with CogVideoX (Yang et al., 2024c) (5B) and DynamiCrafter (Xing et al., 2024) (1.6B) taken out-of-the-box for a system-level comparison. We also compare the proposed method with MotionDirector (Zhao et al., 2024b) (1.8B) which uses image, text, and video as conditional

| Ours *vs.* | Baseline Condition | Domain Finetune | SSv2 | | | Epic Kitchens 100 | | | Fractal | | |
|---|---|---|---|---|---|---|---|---|---|---|---|
| | | | VQ | AT | FC | VQ | AT | FC | VQ | AT | FC |
| WALT (Gupta et al., 2024) | image+text | ✓ | 0.71 | 0.74 | 0.68 | 0.63 | 0.75 | 0.61 | 0.82 | 0.81 | 0.77 |
| MotionDirector (Zhao et al., 2024b) | image+text+video | test-time | 0.86 | 0.95 | 0.89 | 0.97 | 0.99 | 0.96 | 0.95 | 1.00 | 0.96 |
| CogVideoX (Yang et al., 2024c) | image+text | ✗ | 0.78 | 0.89 | 0.86 | 0.78 | 0.89 | 0.81 | 0.96 | 0.99 | 0.99 |
| DynamiCrafter (Xing et al., 2024) | image+text | ✗ | 0.89 | 0.96 | 0.93 | 0.70 | 0.79 | 0.76 | 1.00 | 1.00 | 0.99 |

Table 1: **Human evaluation preference rate on δ-Diffusion.** We compare against four baseline methods in condition video generation that are relevant our task. We ask human raters to evaluate the performance in terms of visual quality (VQ), action transferability between demonstration and generated videos (AT), and initial frame consistency (FC). Note that WALT is fully finetuned on each target domain, MotionDirector optimizes per-instance at test time, and CogVideoX / DynamiCrafter are evaluated out-of-the-box without any finetuning.

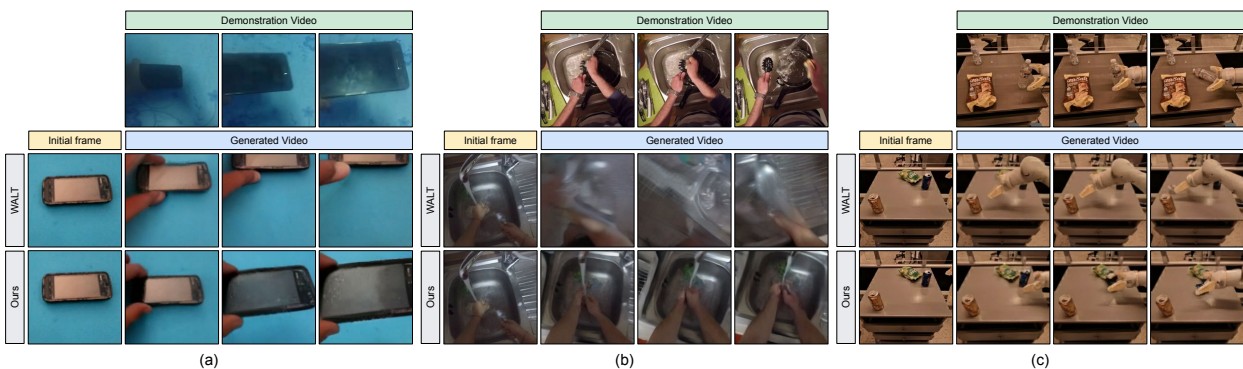

Figure 3: **Qualitative comparisons** of δ-Diffusion against WALT (Gupta et al., 2024) on (a) Something-Something v2 (Goyal et al., 2017), (b) Epic Kitchens 100 (Damen et al., 2018), and (c) Fractal (Brohan et al., 2022) datasets.

signals. When generating videos, MotionDirector learns per-instance appearance and motion from initial frame and demonstration via LoRA fine-tuning (Hu et al., 2022), and generates videos based on the given text prompt. For all baselines, we use the groundtruth captions as the prompt during video generation. Additional details are found in supplementary.

**Human evaluation.** First, we measure human preference on our proposed task. In this study, we hire ten human raters who are trained to assess the overall *visual quality (VQ)*, *action transferability (AT)* and *initial frame consistency (FC)* of the generated videos. The exact rubrics are found in the supplementary. We curate 20 video examples from each dataset, totaling 60 examples for the evaluation set. In each turn, an initial frame, a demonstration video, and two generated videos (one from δ-Diffusion and the other from a baseline) are provided to the rater. The rater is then asked to choose the better video from the two generated ones based on each of the three rubrics mentioned above. We then compute the preference rate of human rater favoring our method against every baseline under each dataset and rubric as the evaluation metric.

**Machine evaluation.** In addition, we apply three quantitative machine metrics to evaluate the action concept transfer quality in different aspects. We first use *Fréchet Video Distance (FVD)* (Unterthiner et al., 2018) score to measure the overall generation fidelity. However, FVD only reflects the distribution shift between generated and reference videos and cannot measure whether the generated video faithfully follow action concept of reference video. To complement this, we measure the similarity between a generated video and the reference video containing the initial frame used in generation via average *embedding cosine similarity (ES)* between the I3D (Carreira & Zisserman, 2017) features of the generated and reference videos. Finally, we construct a retrieval-based evaluation to measure the similarity between the generated and reference video relative to the dataset. Specially, we query with the generated video and attempt to retrieve its corresponding

| Bottleneck | SSv2 | | | Epic Kitchens 100 | | | Fractal | | |
| --- | --- | --- | --- | --- | --- | --- | --- | --- | --- |
| | FVD (↓) | ES (↑) | Hit@100/500/1k (↑) | FVD (↓) | ES (↑) | Hit@100/500/1k (↑) | FVD (↓) | ES (↑) | Hit@100/500/1k (↑) |
| None | 47.3 | 0.838 | 57.7 / 74.3 / 81.7 | 46.2 | 0.847 | 38.9 / 58.9 / 69.1 | **41.9** | 0.902 | 58.0 / 75.7 / 82.8 |
| Temp. Norm. | 38.4 | 0.846 | 63.4 / 79.1 / 85.4 | **41.7** | 0.850 | 41.7 / 61.3 / 70.7 | 42.1 | 0.906 | 62.2 / 78.2 / 85.0 |
| Ours | **38.0** | **0.853** | **66.9 / 81.8 / 87.3** | 42.3 | **0.854** | **44.9 / 64.6 / 74.2** | 44.9 | **0.907** | **63.0 / 78.8 / 85.2** |

Table 2: **Ablation study on $\delta$-Diffusion appearance bottleneck design.** We compare the proposed appearance bottleneck and alternative variants that serve as competitive baselines. For the applied bottleneck, "None" indicates no bottleneck is placed while "Temp. Norm." indicates temporal normalization applied to the spatiotemporal features. We evaluate on Something-Something v2, Epic Kitchens 100, and Fractal for comparisons in terms of generation quality (FVD) and context-generation alignment via both embedding cosine similarity (ES) and retrieval Hit@k (%).

reference video from via I3D cosine similarity. Retrieval performance is measured as average *Retrieval Rate (Hit@k)* at 100, 500, and 1k, which correspond to 1%, 5%, and 10% of the retrieval set, respectively.

### 4.3 Comparison with Baselines

We report the human evaluation of $\delta$-Diffusion against four baselines in Table 1. The numbers indicate preference rate of human rater favoring ours against the baselines under each dataset and rubric. We notice that $\delta$-Diffusion is clearly favored by the human raters across all the datasets and rubrics, underlining its superior performance.

We compare the generated videos by our methods and WALT in Figure 3. WALT, although preserves content consistency well, could carry out in-genuine action to the demonstration video. This could due to the limited expressiveness in text description on demonstration videos. For our results, we show both faithful action execution and content preservation in the generated videos.

### 4.4 Ablation Study

We now ablate the design choice of our training algorithm. We use large scale evaluation data as described in Section 4.1 and machine metrics as described in Section 4.2 for the ablation study. The results are presented in Table 2. We compare two variant bottleneck designs with our proposed appearance bottleneck. "None" stands for directly using the spatiotemporal features $\mathbf{z}^{ST}$ in place of $\delta_V$ without any bottlenecks. For temporal normalization ("Temp. Norm."), following Xiao et al. (2024), the condition signal is obtained by firstly averaging along the temporal dimension of $\mathbf{z}^{ST}$ and then subtracting it from $\mathbf{z}^{ST}$.

We observe that $\delta$-Diffusion performs better than the two counter-parts. On instance-wise metrics, retrieval hit-rate (Hit@k) and embedding cosine similarity (ES), $\delta$-Diffusion consistently outperforms baselines with none or temporal normalization as the bottleneck. This indicates an improved action concept transferability with our proposed appearance bottleneck design. On the other hand, we do not observe consistent win among the three methods in terms of visual quality. This shows that while increasing the action faithfulness and temporal con-

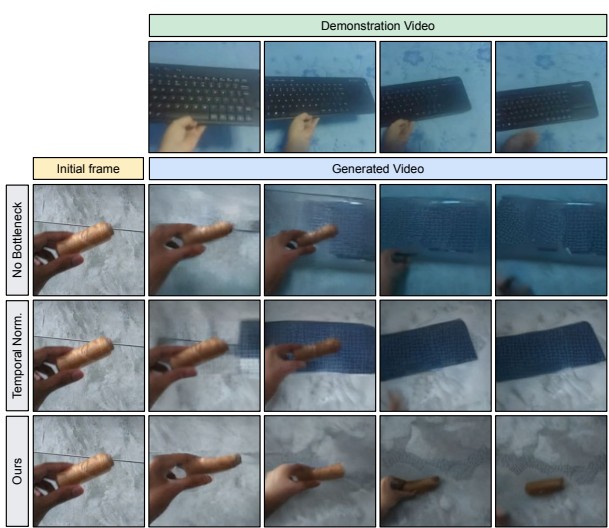

Figure 4: **Qualitative results for bottleneck ablation** on the Something-Something v2 dataset (Goyal et al., 2017). Applying no or temporal normalization bottleneck suffers from appearance leakage, while our appearance bottleneck preserves the input context consistently.

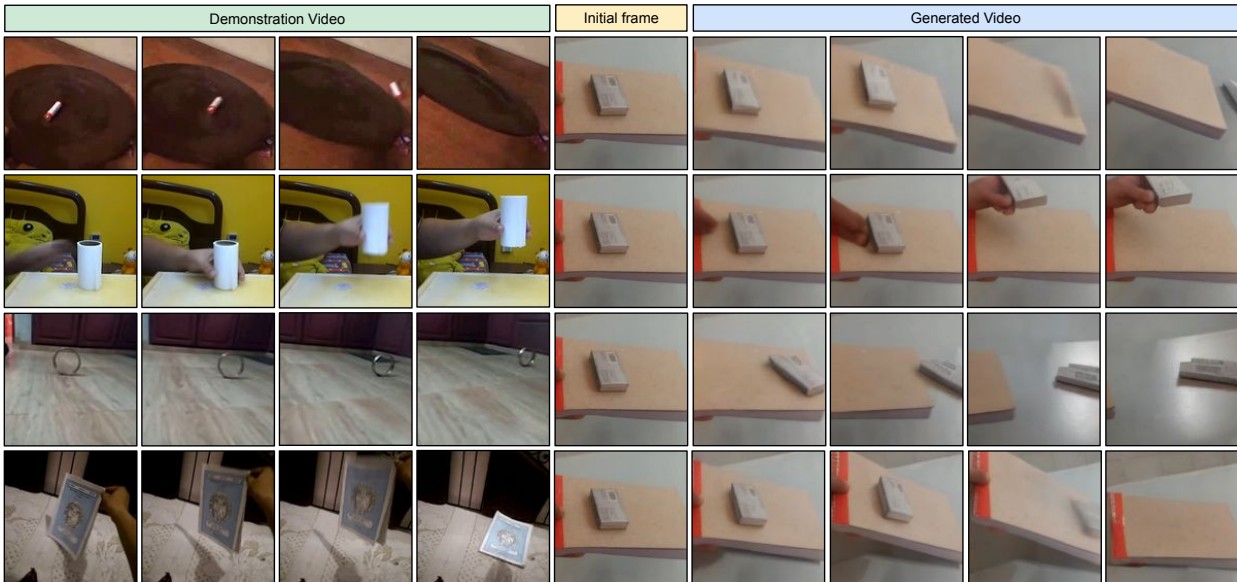

Figure 5: **Controllability by demonstration.** Qualitative results of driving alternative generation from the same initial frame with different demonstration videos from the Something-Something v2 dataset (Goyal et al., 2017).

sistency, the proposed method does not hurt the visual quality of the generated videos. Figure 4 further shows visual comparisons for different bottleneck designs. We notice that both without bottleneck and the "temporal norm." bottleneck suffer from appearance leakage. In contrast, the proposed appearance bottleneck successfully retains action concepts from the demonstration video while minimizing the appearance information.

### 4.5 Qualitative Results

**Controllability by demonstration.** As shown in Figure 5, $\delta$-Diffusion can generate videos with various actions starting from the same initial frame by conditioning on different demonstration videos. More results can be found in the supplementary materials.

**Compositional generation.** In Figure 6, we further explore generating a coherent video that starts from a single initial frame and conditioned on a sequence of demonstration videos. After each generation step, $\delta$-Diffusion uses the last frames from the previous step as the next initial frame. Each demonstration video is selected from a different scene, but together they showcase a complete rollout of a complex task. We show that by sequentially applying the action latents from the demonstration videos, we are able to auto-regressively simulate the task execution in one consistent environment provided by the initial frame.

## 5 Conclusion

In this work, we introduce Video Creation by Demonstration, a video creation experience that enables users to generate videos by providing an initial frame and a demonstration video for action concept conditioning. The main challenge in this task is how to handle the misalignment of appearances and contexts between the demonstration video and the initial frame during the transferring of action concepts. To address this, we propose to leverage the video foundation model to provide semantic representations for actions and contexts in videos and build an appearance bottleneck on top of them to extract latent control signals with rich action concept and minimum appearance information. Then, a video generation model is learned to take the initial frame and the control signals as inputs and generate videos that continue from the initial frame and carry out

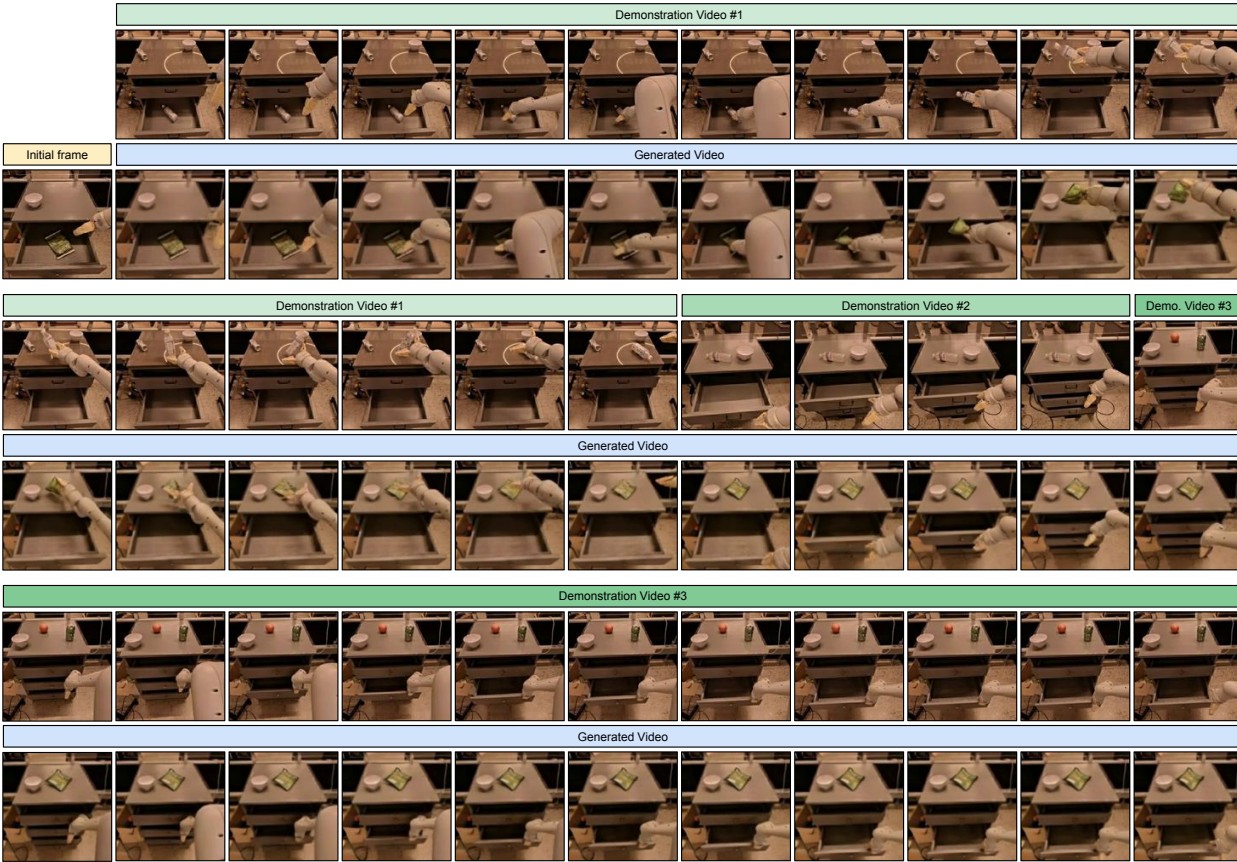

Figure 6: **Compositional generation** controlled via a concatenation of three different demonstration videos of varying lengths. The sequence of demonstrated action concepts ("picking something from a drawer and placing it on the table", "closing a drawer", and "opening a drawer") are coherently rollout in the environment provided by the input initial frame.

the action concept in the similar manner as shown in the demonstration video. Our extensive experiments on three separate datasets using both machine and human evaluations demonstrate the effectiveness of our proposed δ-Diffusion. One of the limitations is that δ-Diffusion at its current form does not always strictly preserve physical realism under complex scenes. This would require further exploration and may be potentially alleviated by scaling up the generation model capacity.

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

# A   Appendix

## A.1   Additional Implementation Details

**Vision encoders.**   We adopt the video foundation model VideoPrism (Zhao et al., 2024a) (Base, 0.1B parameters) as the vision encoders $\mathcal{F}$. VideoPrism first computes the spatial representations per frame and then temporally aggregates each spatial location to output the final spatiotemporal representations. This factorization in space and time allows the appearance bottleneck to directly take the per-frame spatial representations in VideoPrism as input, which reduces both training and inference cost. VideoPrism takes $16$ $288 \times 288$ frames as input and outputs a $16 \times 16 \times 16 \times 768$ spatiotemporal representation tensor.

**Appearance bottleneck feature predictor.**   The per-frame feature predictor $\mathcal{P}$ is constructed as a stack of 4 Transformer encoder blocks from ViViT (Arnab et al., 2021) with 1024 hidden dimensions and 8 heads for multi-head attention. $\mathcal{P}$ takes the intermediate VideoPrism spatial encoder output for a particular frame $V_t$ as input, and predicts its associated frame representations $\mathbf{z}_t^{\mathrm{ST}}$. During training, we adopt a reconstruction objective with L2 loss and trained for 30k iterations with the Adam optimizer (Kingma, 2015) with $10^{-4}$ base learning rate and $4.5 \times 10^{-2}$ weight decay.

**Video generation model.**   For the video generation model $\mathcal{G}$, we adopt WALT (Gupta et al., 2024) (Large, 0.3B parameters). $\mathcal{G}$ is trained to generate videos of $T = 16$ frames with $128 \times 128$ spatial resolution. The initial frame $I$ is natively passed into the architecture as the image for conditional generation. The control latents $\delta_V$ from demonstration video $V$ is projected into a sequence of 2048-dimensional latent vectors in place of the original text embeddings for conditioning.

## A.2   Improving Generation Quality

To support demonstration video with length longer than the generation length $T = 16$, we follow the auto-regressive generation setup in WALT (Gupta et al., 2024). During model $\mathcal{G}$ training, we swap out the input initial frame with a probability of 0.5 and replace it with multiple consecutive frames to encourage a smoother continual generation. At inference time, we cut long demonstration video into multiple segments as needed, each of length $T$. This allows the generated segment to remain aligned with the appropriate demonstration segment. Here, the first segment is generated from the input initial frame, while the subsequent segments are conditioned on the last 4 generated frames from a previous segment.

During training, we enable self-conditioning (Chen et al., 2023a) with a probability of 0.9 and randomly mask out the control signal with a probability of 0.2. At inference time, we adopt classifier-free guidance consistent with Saharia et al. (2022) with a guidance weight of 1.25 and drop both the self-conditioning and control signal for the unconditional generation.

## A.3   Human Evaluation Setup

### A.3.1   Prompt Selection

For all three datasets, we consider the demonstration video to be between 16 to 24 frames at 12 frames per second (FPS) (Something-Something v2 and Epic Kitchens 100) or 10 FPS (Fractal). For Something-Something v2 (Goyal et al., 2017), we select the top-10 action classes. For each action class, we manually select 20 out of 500 randomly sampled pairs for human evaluation. For Epic Kitchens 100 (Damen et al., 2018), we apply automatic filtering to first narrow down the search. For each video, we retrieve top-1 video with the same action label and different participant ID via first-frame CLIP (Radford et al., 2021) similarity. Then, 20 pairs are randomly sampled from top-25 action classes. Finally, 20 examples are manually selected from a pool of pairs with CLIP similarity greater than 0.9. For Fractal, we sample 500 same-action pairs uniformly from each of the 8 action classes (defined by the verb and preposition from the associated captions) and conduct manual selection in random order to select 20 pairs for evaluation. The main manual selection principle is to confirm that the action concept in the demonstration video can be potentially applied in a

| Instructions |
| --- |
| *Given an image (initial frame) and a demonstration video (reference video), the task of Video Creation by Demonstration is defined as to create a plausible video clip initiating from the initial frame and contains similar content dynamics as in the demonstration video.* |
| *In this user study, you will be provided a reference video, an initial frame, and two generated videos from two methods side-by-side in each turn. You would assess the generated video quality on the following three rubrics and determine which one better recreates the actions and dynamics of a given demonstration video, while also appearing realistic and continuous from an initial frame.* |

| Rubrics | Descriptions |
| --- | --- |
| Visual Quality | *How realistic does the video look compared to a real-world video? Consider factors like smooth motion, accurate details, and believable physics. Choose the video that is better.* |
| Action Transferability | *How well does the generated video recreate the actions and movements shown in the reference video? Note that the action concept can include the camera motions. Choose the video that is better.* |
| Frame Continuity | *How seamlessly does the generated video flow from the provided initial frame? Does it look like a natural continuation of the scene? Choose the video that is better.* |

Table 3: Human evaluation instructions and rubrics.

reasonable way from the initial frame. This process is conducted without any considerations of the methods to be evaluated.

From each selected video pair, one video is randomly selected as demonstration, and the first of frame of the other video is used as the initial frame to construct a prompt.

### A.3.2 Rating Instructions and Rubrics

In Table 3, we list the instructions and rubrics shown to the human raters before presenting them with the generated videos for side-by-side comparisons.

### A.3.3 Additional Details

When presenting the visual examples, we show **Demonstration Video**, **initial frame**, **Video A**, and **Video B** in order. As we conduct the user study via Colab, we randomly assign *Video A* to be either $\delta$-Diffusion or a baseline method. We hired 10 human raters to examine 4 baselines to compare against $\delta$-Diffusion on 3 datasets, 20 examples per dataset, and 3 metrics per example. In total, we collected 7200 human preferences.

### A.4 Details on Baselines

**MotionDirector (Zhao et al., 2024b).** When testing MotionDirector (1.8B) on Something-Something v2 (SSv2) (Goyal et al., 2017), Epic Kitchens 100 (Damen et al., 2018), and Fractal (Brohan et al., 2022), we use the official code published by the authors at `https://github.com/showlab/MotionDirector`. We first train the spatial path with the initial frame for 300 epochs with input resolution being $384 \times 384$. Then we train the temporal path following the 16-frame single video setup. During inference, a noise prior of 0.0 is applied. Ground truth caption labels are used as the prompt for both training and inference.

**WALT (Gupta et al., 2024).** We fine-tune WALT-L (0.3B) on each dataset in accordance with the official procedures and use ground truth captions to tune video generation from initial frame and text caption. We keep all hyper-parameters the same and fine-tune for 200k steps. During inference, the ground truth caption of the demonstration video is used along with the initial frame for video generation. As shown in the visualizations, WALT generations, while preserving initial frame consistency, carry out ungenuine action due to limited expressiveness in text descriptions.

**CogVideoX (Yang et al., 2024c) and DynamiCrafter (Xing et al., 2024).** We take both method out-of-the-box and evaluate with no finetuning. Note that although not finetuned, both CogVideoX (5B) and DynamiCrafter(1.6B) are much larger than $\delta$-Diffusion (0.3B) in terms of model capacity.

### A.5 Qualitative Results

The original videos for creating Figures 1 and 6 can be found at `index.html` in the supplementary materials. The original videos for creating Figures 4 and 3 can be also found along with additional examples.

### A.6 Controllability by Demonstration

As shown in Figures 8, 9, 10, and 11, we showcase the controllability of $\delta$-Diffusion using different demonstration videos for the same initial frame. The video samples can be found at `index.html/#cbd` in the supplementary materials.

### A.7 Failure Cases

As shown in Figure 7, we identify three primary failure modes of our method. In the first row, we show a case where the semantics of the action concept in the demonstration videos are not fully carried out. Specifically, the generated object is placed "in-between" the existing objects instead of "next-to" them. In the second row, we show a case where permanence is not held when the object in the demonstration video undergoes fast appearance changes. Here, fast object rotation causes appearance leakage in the generation. In the third row, we show inconsistent generations where the demonstration videos and initial frames are mis-matched significantly. On the left, the perspectives of the moving hand are mis-matched and cause another hand in the same demonstrated perspective to be generated.

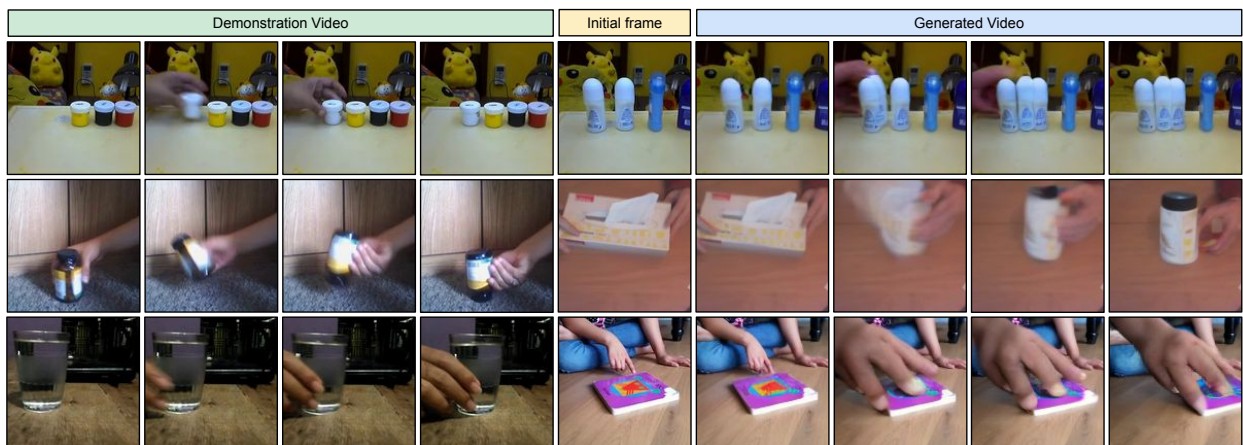

Figure 7: Failure cases generated by $\delta$-Diffusion.

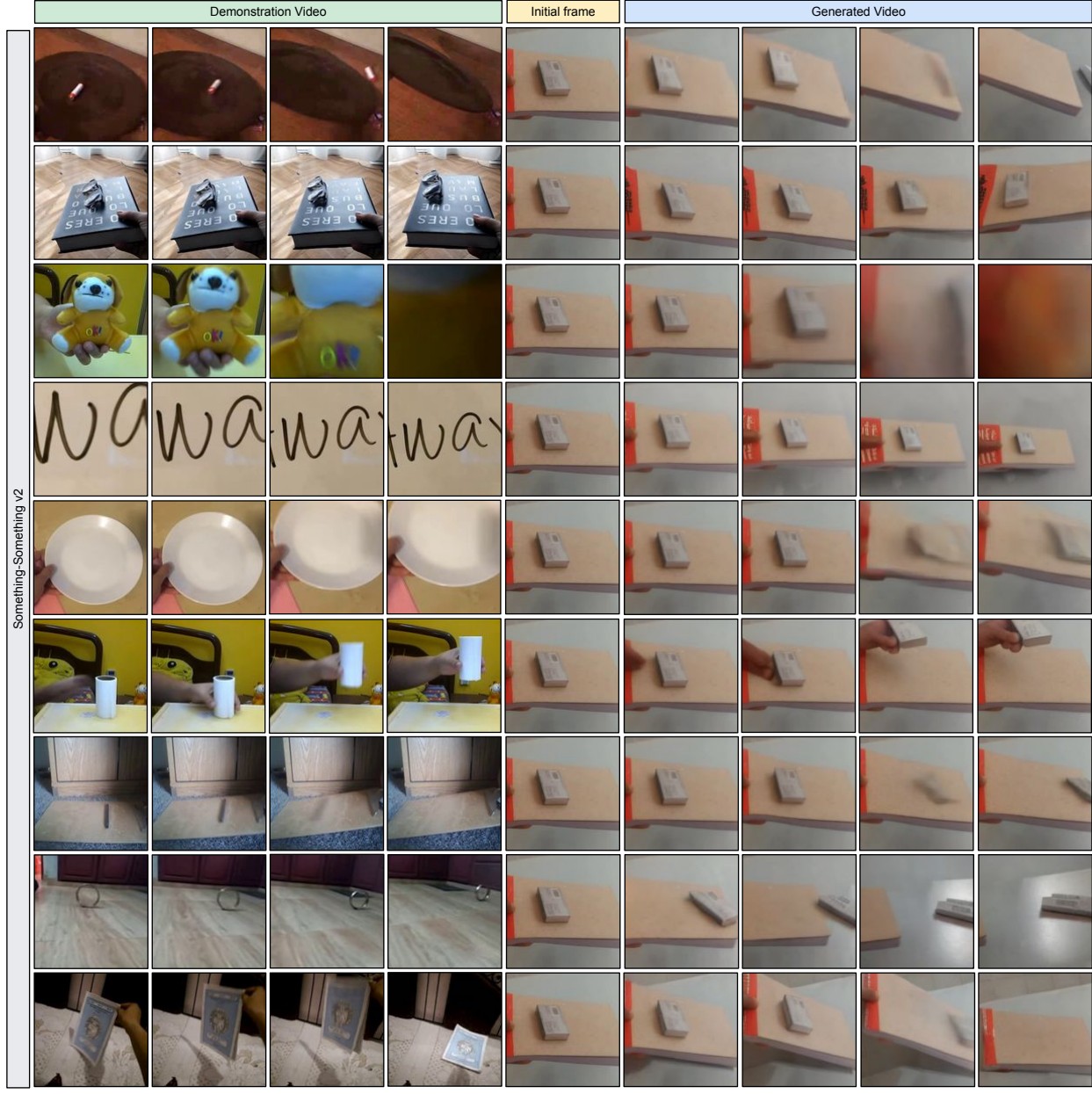

Figure 8: Qualitative results of driving alternative generation from the same initial frame with different demonstration videos from the Something-Something v2 dataset (Goyal et al., 2017).

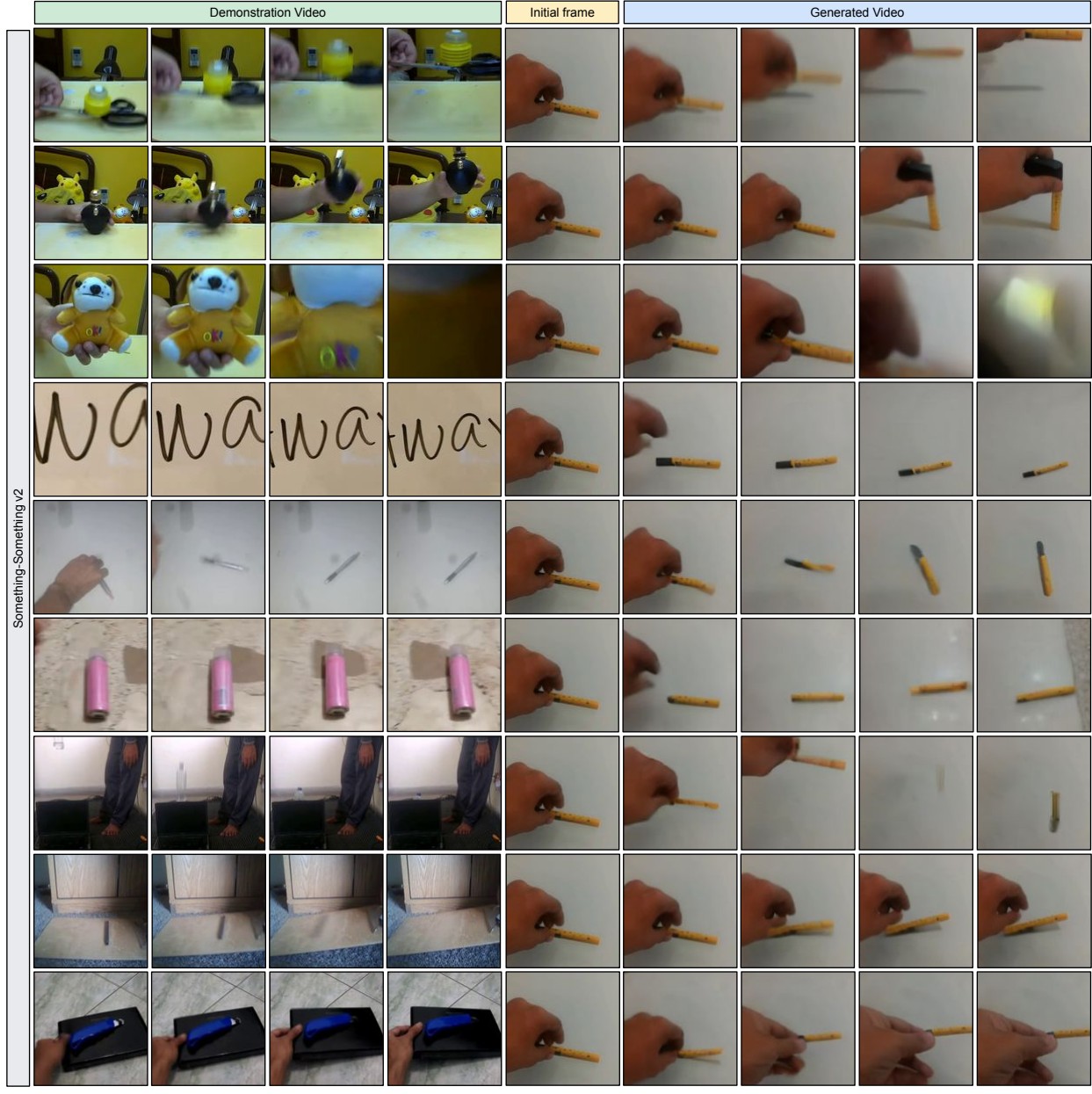

Figure 9: Qualitative results of driving alternative generation from the same initial frame with different demonstration videos from the Something-Something v2 dataset (Goyal et al., 2017).

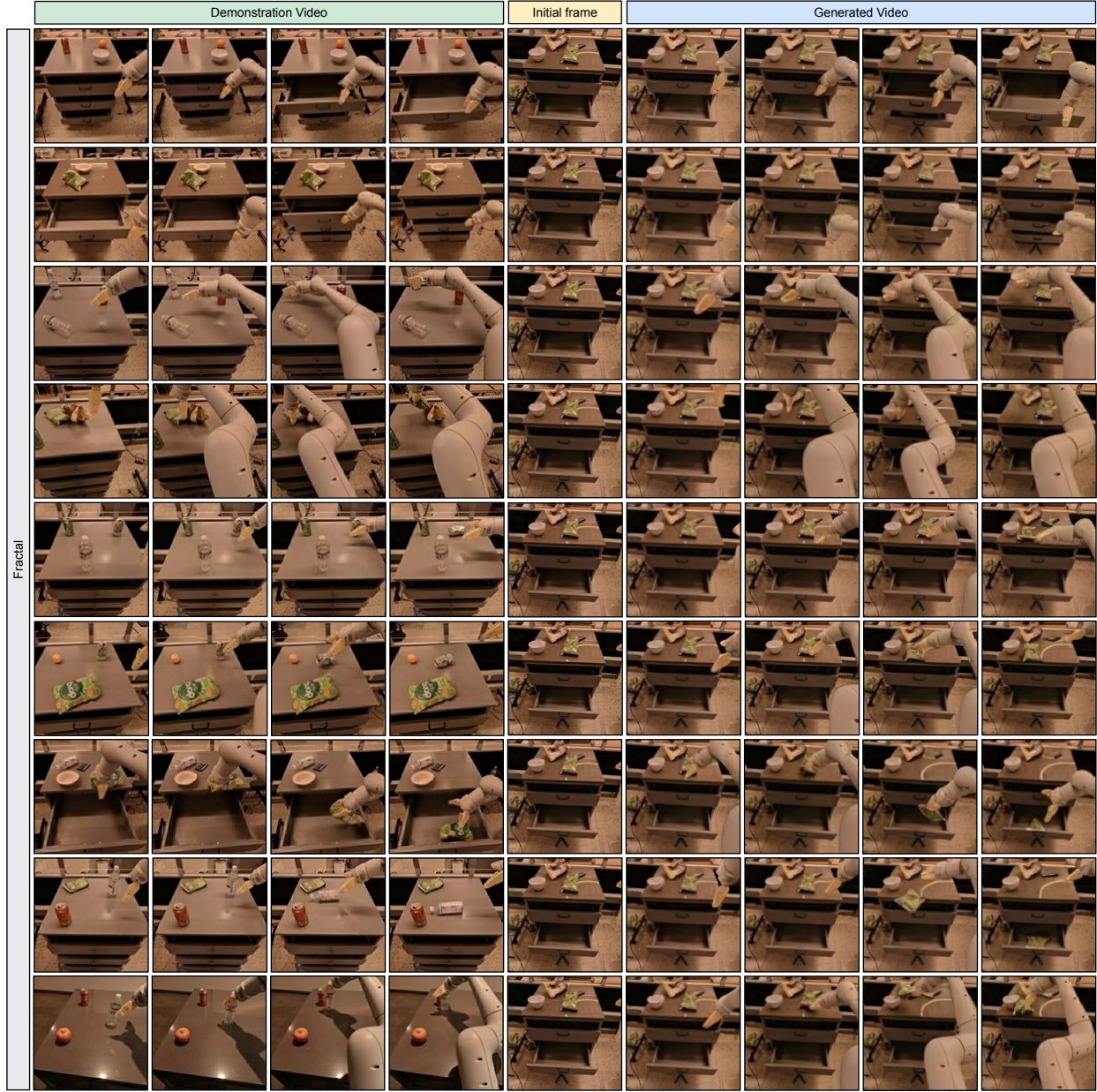

Figure 10: Qualitative results of driving alternative generation from the same initial frame with different demonstration videos from the Fractal dataset (Brohan et al., 2022).

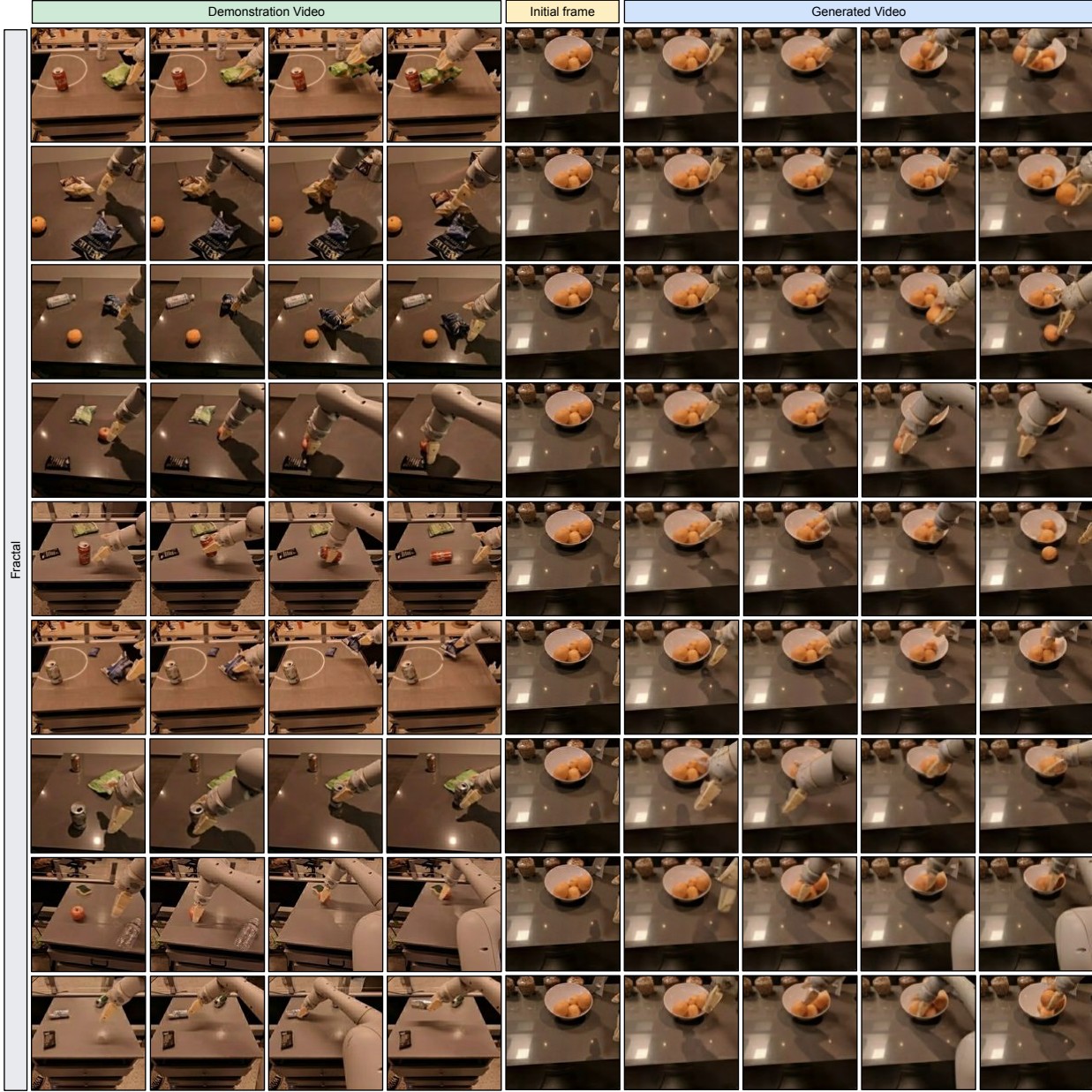

Figure 11: Qualitative results of driving alternative generation from the same initial frame with different demonstration videos from the Fractal dataset (Brohan et al., 2022).

