# OpenReview forum: "Video Creation by Demonstration"
_TMLR — Under review for TMLR_

### Review · Reviewer_wRqH · 2025-11-04

**Summary Of Contributions:**

This paper presents a novel approach called δ-Diffusion for generating videos from a single frame but that follow the action contained in another video. The method is fully self-supervised and does not require any curated pair of videos following the same action. The paper presents convincing qualitative and qualitative evaluation, including human evaluation.

**Audience:**

Yes

**Audience Explanation:**

This paper is at the intersection of several research topics and would be of great interest for their communities:

- Self-supervised learning. One interesting component of the proposed method is that it is fully self-supervised, and in particular from video inputs, which is a highly active research topic.

- Video generation. Obviously people interested in the problem of conditional generation on instructional video.

- Latent action models / robotics. This research topic is new and still emerging, the objective is to learn a common action space for various embodiments, potentially from large-scale internet natural video.

**Claims And Evidence:**

Yes

**Claims Explanation:**

Strengths:

- The proposed idea is simple, the paper is clearly written, easy to follow, and has a nice presentation.

- The experimental results, and in particular the qualitative results, show that the approach is working. Human evaluations confirm that this is not only due to cherry picked examples.

Weaknesses:

- The paper does not describe precisely what is in the training data. First in section 4.1, some datasets are mentioned: Epic Kitchens 100, SSv2, Fractal, but it is unclear if the training data is the union of these datasets or something else, then section 4.2 mentions “We additionally include Ego4D” which makes things even more confusing. The composition of the training data (diversity, scale) is crucial to understand the effectiveness of the approach. Indeed, if the model is already trained on a very-large scale dataset, that gives very few room for improvement and scaling. Right now despite the qualitative results being impressive, I don’t have a sense of whether this method would scale, and it makes it hard to be fully convinced by the approach.

- The paper does not make strong connections between the proposed approach and the Latent Action Models (LAM) literature. Dynamo and LAPA are briefly mentioned in section 2.2 but that’s it. In particular in terms of motivation, I think the scope of this paper goes beyond video generation, which is a nice problem, but not as interesting as the problem of learning a universal latent space for actions and embodiments, which has applications in robotics and control. I find it underwhelming that the paper only talks and focuses on video generation.

- Qualitative evaluation is convincing but prone to cherry-picking. Human evaluation fixes this issue and shows the superiority of the approach, however I am not really convinced by the non-human machine evaluations. Even if it is acknowledged by the authors, I think that FVD is highly uninformative for this task, and cosine similarity uses a single reference video while there is some uncertainty in the prediction and there are several possible continuation to an input frame.

- The “contributions” paragraph at the end of the introduction sounds weird because: (i) and (ii) and (iii) are essentially the same contribution. I think separating in 3 contributions is irrelevant here and it is ok to have a single important contribution.


Questions:

- Is the Per-frame Spatial Encoder trained at the same time as the Predictor ? In this case could there be collapsing solutions where the output of the predictor is 0 and delta_V has the same information as Z_ST.

- Have you tried to condition on both text and a video ? For example the text could describe additional information that is not about the action but about visual aspects that would change from the reference frame.

**Requested Changes:**

The main changes are:
- Clarify the training data.
- Make connections to the recent Latent Action Model literature.

Otherwise the paper is of good quality and presents interesting findings for the TMLR community.

---

> ### Author Response · Authors · 2026-07-08
>
> We thank the reviewer for the detailed and constructive feedback. We address each point below.
>
> ### **Training Data Composition**
>
> We apologize for the ambiguity. To clarify: the appearance bottleneck (feature predictor $\mathcal{P}$) is trained on the union of EK100, SSv2, Fractal, and Ego4D. The diffusion model $\mathcal{G}$ is then trained separately on each individual dataset (EK100, SSv2, or Fractal), with $\mathcal{P}$ frozen. We include Ego4D only during the bottleneck training stage due to its rich and diversified content, which helps $\mathcal{P}$ generalize across domains. We will clarify this.
>
> ### **Connections to the Latent Action Model Literature**
> We agree that stronger connections to the Latent Action Model (LAM) literature would better situate our work, and we will expand the related work discussion accordingly. Like LAM works, our appearance bottleneck learns an implicit action representation from unlabeled video without requiring action annotations. Although we showcase its efficacy in video generation, we believe that the broader applicability of our approach — e.g., as a general-purpose latent action space for embodied agents — is an exciting future direction.
>
> ### **Machine Evaluation Metrics**
>
> We agree that FVD is an imperfect metric for this task, and that cosine similarity against a single reference video does not fully account for the diversity of possible continuations. This is why we place greater emphasis on the human evaluation, which directly measures visual quality, action transferability, and frame consistency.
>
> The machine evaluation is intended to complement this with broader coverage. Crucially, since the reference video and demonstration video share the same action label, the set of plausible continuations from the initial frame is constrained by the underlying action category, which partially mitigates the multi-modality concern for the cosine similarity metric.
>
> ### **Contributions Paragraph**
>
> We thank the reviewer for this observation. We will revise the contributions paragraph accordingly.
>
> ### **Collapsing Solutions in the Appearance Bottleneck**
>
> In practice, the per-frame spatial encoder is frozen throughout training; only the predictor $\mathcal{P}$ is trained. Since $\mathcal{P}$ is optimized to *minimize* $\delta_V$ (i.e., to approximate $\mathbf{z}_t^{\mathrm{ST}}$ as closely as possible from $\mathbf{z}_t^{\mathrm{S}}$), the degenerate solution where $\mathcal{P}$ outputs zero would not occur in practice.
>
> We note, however, that a different collapsing solution *can* arise if $\mathcal{G}$ and the appearance bottleneck are trained jointly: the reconstruction objective would then incentivize $\delta_V$ to retain full appearance information, potentially driving $\mathcal{P}$ toward near-zero outputs. This is precisely why we adopt a two-stage training paradigm where $\mathcal{P}$ is frozen before $\mathcal{G}$ is trained.
>
> ### **Joint Conditioning on Text and Video**
>
> We consider joint text-and-video conditioning an interesting direction for future work. In this paper, we specifically focus on the setting where videos alone drive generation and can be learned without supervision, as we highlight the expressiveness of video as a control signal.

---

### Review · Reviewer_Fdo9 · 2025-11-06

**Summary Of Contributions:**

The paper proposes Video Creation by Demonstration, a method that generates realistic videos continuing from an initial frame while replicating the action shown in a demonstration video.
Unlike text or pose-based controls, this approach uses videos to convey detailed action information without manual annotations.
To overcome the lack of paired supervision, the authors introduce $\delta$-Diffusion, a self-supervised diffusion framework that employs a feature bottleneck to isolate action representations from video foundation models.
The method achieves superior performance compared to existing baselines in both human and quantitative evaluations.

**Audience:**

Yes

**Audience Explanation:**

The work shows a new way of generating image-based videos following the moving action from a reference video.
Despite a quite specific task, the method is meaningful for the research and industrial community.

**Broader Impact Concerns:**

No impact concerns.

**Claims And Evidence:**

Yes

**Claims Explanation:**

The paper delivers a new method that can generate videos from an initial frame that mimic the action from a reference video.
The novelty of the paper lies in the leverage of a disentangle of action information by learning the difference of spatial features and spatiotemporal features from a pre-trained video foundation model.
The results are impressive though some limitations remain.

**Requested Changes:**

- While the appearance bottleneck is conceptually elegant, its theoretical justification (e.g., what precise aspects of action semantics it isolates) is primarily empirical. The work would benefit from deeper analysis or formalization of this representation. If no, could this concept of action semantics be generalize to other video generation models or tasks?

- Following the first issue, $\delta$-Diffusion’s success heavily relies on the representational quality of the underlying video foundation encoder (e.g., VideoPrism). The approach’s adaptability to weaker or domain-specific encoders is not explored.

- The model generates relatively short and low-resolution clips (16 frames at 128×128), which limits the demonstrated realism and generalization to long-horizon or high-definition scenarios. Could this method inference on longer videos with complex movement?

- Most baselines are text- or prompt-based diffusion models. However, the proposed method has the access temporal information richer than prompt. If possible, the paper could be strengthened by including comparisons with other unsupervised, representation-driven or reference-video-based generative frameworks.

Because of its impressive quality and method, I recommend an acceptance of the manuscript after revision.

---

> ### Author Response · Authors · 2026-07-08
>
> We thank the reviewer for the positive recommendation and constructive feedback. We address each point below.
>
> ### **Theoretical Justification of the Appearance Bottleneck**
> We agree that a more formal justification would strengthen the paper and will include the following analysis in the revision.
>
> The core intuition behind the appearance bottleneck is that spatiotemporal features from a video encoder contain a mixture of appearance information (what objects look like in a given frame) and action information (how those objects move over time). To isolate the action signal, we need to subtract out the appearance component. We formalize this as follows.
>
> The bottleneck rests on two assumptions. First, the spatiotemporal feature $\mathbf{z}_t^{\mathrm{ST}}$ for frame $V_t$ admits an additive decomposition: $$\mathbf{z}_t^{\mathrm{ST}} = \mathbf{z}_t^{\mathrm{APP}} + \mathbf{z}_t^{\mathrm{ACT}},$$ where $\mathbf{z}_t^{\mathrm{APP}}$ is an *appearance component* determined solely by the per-frame spatial feature $\mathbf{z}_t^{\mathrm{S}}$, and $\mathbf{z}_t^{\mathrm{ACT}}$ is an *action component* arising from the temporal context of the full clip. This decomposition is motivated by the factorized spatial-then-temporal structure of VideoPrism, where temporal aggregation operates on top of independently computed per-frame spatial features.
>
> Second, we assume that action information is on average independent of per-frame appearance, i.e., $$\mathbb{E}[\mathbf{z}_t^{\mathrm{ACT}}] = c \in \mathbb{R}^d,$$ for some global constant $c$ capturing dataset-level biases. This assumption is reasonable because the action being performed in a clip (e.g., pushing, lifting) is not, in expectation, predictable from a single frame alone. Under these two assumptions, the conditional expectation of $\mathbf{z}_t^{\mathrm{ST}}$ given $\mathbf{z}_t^{\mathrm{S}}$ satisfies: $$\mathbb{E}[\mathbf{z}_t^{\mathrm{ST}} \mid \mathbf{z}_t^{\mathrm{S}}] = \mathbb{E}[\mathbf{z}_t^{\mathrm{APP}} \mid \mathbf{z}_t^{\mathrm{S}}] + \mathbb{E}[\mathbf{z}_t^{\mathrm{ACT}} \mid \mathbf{z}_t^{\mathrm{S}}] \approx \mathbf{z}_t^{\mathrm{APP}} + c,$$ since $\mathbf{z}_t^{\mathrm{APP}}$ is a deterministic function of $\mathbf{z}_t^{\mathrm{S}}$, and $\mathbb{E}[\mathbf{z}_t^{\mathrm{ACT}} \mid \mathbf{z}_t^{\mathrm{S}}] \approx c$ under the independence assumption.
>
> Subtracting this conditional expectation from $\mathbf{z}_t^{\mathrm{ST}}$ then yields the action embedding: $$\delta_t = \mathbf{z}_t^{\mathrm{ST}} - \mathbb{E}[\mathbf{z}_t^{\mathrm{ST}} \mid \mathbf{z}_t^{\mathrm{S}}] \approx \mathbf{z}_t^{\mathrm{ACT}} - c,$$ which retains action information while suppressing per-frame appearance. In practice, we approximate the intractable conditional expectation $\mathbb{E}[\mathbf{z}_t^{\mathrm{ST}} \mid \mathbf{z}_t^{\mathrm{S}}]$ with a learned predictor $\mathcal{P}(\mathbf{z}_t^{\mathrm{S}})$, trained via the L2 objective in Eq. (1) of the main paper. This recovers $\delta_V = \mathbf{z}_t^{\mathrm{ST}} - \mathcal{P}(\mathbf{z}_t^{\mathrm{S}})$ as defined in the main method.
>
> ### **Reliance on VideoPrism**
>
> We acknowledge that a full encoder ablation is beyond the scope of this revision. However, the reliance on VideoPrism is not fundamental: the appearance bottleneck requires only that the encoder factorizes spatial and temporal features independently, a design shared by a broad class of modern video foundation models. We therefore expect performance to scale naturally with encoder quality, and we will make this architectural requirement explicit in the revised paper.
>
> ### **Short and Low-Resolution Generation**
>
> The core contribution of this paper is the action transfer mechanism — the appearance bottleneck and self-supervised training paradigm — rather than a state-of-the-art video generation system. Resolution and clip length are properties of the underlying generation model (WALT), and scaling them is orthogonal to our central claims. That said, our compositional generation already supports longer sequences via autoregressive chaining of segments; we will make this connection more prominent in the main paper.
>
> ### **Comparison Against Reference-Video-Based Frameworks**
> MotionDirector is a reference-video-driven baseline: it takes a demonstration video as input to drive generation, and δ-Diffusion outperforms it substantially across all three datasets and all three human evaluation rubrics. We believe this comparison already addresses the existing concern, and will make this point more explicit in the discussion.

---

### Review · Reviewer_Tzrr · 2026-06-23

**Summary Of Contributions:**

This paper proposes a method that generates videos starting from a given initial frame while replicating actions from a demonstration video. The key contribution is δ-Diffusion, a self-supervised diffusion-based model that extracts action representations from demonstration videos, ensuring that the generated video maintains both realistic motion and fidelity to the initial frame. The method leverages pre-trained video foundation models, requires no paired supervised training data. Additional contributions include compositional generation by chaining multiple demonstration videos and qualitative insights into failure modes.

**Audience:**

Yes

**Audience Explanation:**

The work addresses controllable video generation, a highly relevant topic in machine learning and multimedia research.

**Claims And Evidence:**

Yes

**Claims Explanation:**

The authors provide extensive quantitative and qualitative evidence. Human evaluation (preference rates on visual quality, action transferability, and frame consistency) shows δ-Diffusion consistently outperforms strong baselines such as WALT, MotionDirector, CogVideoX, and DynamiCrafter. Ablation studies validate the effectiveness of the appearance bottleneck, showing it reduces appearance leakage and improves alignment between action and context.

**Requested Changes:**

1. The authors note δ-diffusion does not always preserve physical realism under complex scenes. Recommend discussing potential solutions, e.g., scaling model capacity or incorporating physics constraints. For the showing case in the demo webpage (## Qualitative Results Something-Something v2, first row and second column), if the cup is replaced with a bottle with water and the slope angle of the black book in the demonstration video is increased, will water spill out of the cup? Another interesting example, could you use a toy car to drive a real car?
2. Compositional generation currently chains small segments; suggest evaluating how well long sequences maintain coherent action and scene context.
3. Include quantitative analysis on scenarios where the initial frame partially lacks required objects for the demonstrated action.
4. Consider expanding experiments to include non-egocentric or more diverse datasets to validate generalization beyond the current three. If in the demonstration video there are two dynamic objects with conflicting motion, and the context image has one object or multiple (>2) objects, what about the results?
5. Could you provide an image or video matrix, where the first row is the demonstration video and the first column is the different context image with large variance, and then fill out the other entry of the matrix?

---

> ### Author Response · Authors · 2026-07-08
>
> We thank the reviewer for the positive assessment and thoughtful questions. We address each point below.
>
> ### **Physical Realism Under Complex Scenes**
> We do observe scenarios where the model demonstrates correct physical realism. As shown in the supplementary materials, physical collisions are modeled correctly in multiple examples (see `static/videos/main/fractal/{FR-EXT50A,FR-EXT16A,FR-EXT3A}.mp4`). Regarding the water-spilling scenario, we refer the reviewer to `static/videos/main/ek100/EK-EXT147B.mp4`, where noodles inside a container can be poured out by using a demonstration video showing the same motion with an empty bowl. While these examples showcase the physical realism of the proposed method, we note that physical realism is an ongoing direction for video generation research, making it a more orthogonal research question to controlling video generation by demonstration.
>
> ### **Coherence of Long Compositional Sequences**
>
> We consider long compositional generation a promising capability of our model as shown in Fig. 5. However, since our training distribution focuses on short video clips, we do not claim superiority in long-form video generation. Evaluating how well long sequences maintain coherent action and scene context over many chained demonstrations is an interesting future work.
>
> ### **Missing Objects in the Initial Frame**
>
> This is a good point. We consider the task feasible only when the initial frame depicts a scene that can support the target action. In our development, when the initial frame does not contain an appropriate object for the demonstrated action, we observe that the model tends to transfer only the camera motion in the demonstration without any object-level action.
>
> ### **Generalization to More Diverse / Non-Egocentric Datasets & Conflicting Motion**
>
> Our work focuses on transferring action signals from a demonstration video, and we find egocentric datasets best showcase the variability of possible actions to transfer. Thus, we consider extension to non-egocentric settings as an interesting future direction to our work. Regarding conflicting motion between multiple dynamic objects: we consider such cases as effectively infeasible pairings between the demonstration and the initial frame. In practice, when multiple source objects with conflicting motions are paired with a single target object in the initial frame, we observe that the model tends to transfer the demonstrated action of the source object that is spatially closest to the target relative to the frame.
>
> ### **Visualizing different input scenarios**
>
> In the supplementary materials, we showcase multiple examples where different demonstration videos are paired with the same initial frame.